# Unique Salt-Tolerance-Related QTLs, Evolved in *Vigna riukiuensis* (Na^+^ Includer) and *V. nakashimae* (Na^+^ Excluder), Shed Light on the Development of Super-Salt-Tolerant Azuki Bean (*V. angularis*) Cultivars

**DOI:** 10.3390/plants12081680

**Published:** 2023-04-17

**Authors:** Eri Ogiso-Tanaka, Sompong Chankaew, Yutaro Yoshida, Takehisa Isemura, Rusama Marubodee, Alisa Kongjaimun, Akiko Baba-Kasai, Kazutoshi Okuno, Hiroshi Ehara, Norihiko Tomooka

**Affiliations:** 1Genetic Resources Center, National Institute of Agrobiological Sciences (NIAS), 2-1-2 Kannondai, Tsukuba 305-8602, Ibaraki, Japan; 2Program in Plant Breeding, Faculty of Agriculture at Kamphaeng Saen, Kasetsart University, Kamphaeng Saen, Nakhon Pathom 73140, Thailand; 3Graduate School of Life and Environmental Sciences, University of Tsukuba, Tennohdai 1-1-1, Tsukuba 305-8571, Ibaraki, Japan; 4Graduate School of Bioresources, Mie University, 1577 Kurimamachiya-cho, Tsu 514-0102, Mie, Japan

**Keywords:** crop wild relatives, salt tolerance, Na^+^ excluder, Na^+^ includer, genetic resources, marker assisted selection, SSR, RAD-seq

## Abstract

Wild relatives of crops have the potential to improve food crops, especially in terms of improving abiotic stress tolerance. Two closely related wild species of the traditional East Asian legume crops, Azuki bean (*Vigna angularis*), *V. riukiuensis* “Tojinbaka” and *V. nakashimae* “Ukushima” were shown to have much higher levels of salt tolerance than azuki beans. To identify the genomic regions responsible for salt tolerance in “Tojinbaka” and “Ukushima”, three interspecific hybrids were developed: (A) azuki bean cultivar “Kyoto Dainagon” × “Tojinbaka”, (B) “Kyoto Dainagon” × “Ukushima” and (C) “Ukushima” × “Tojinbaka”. Linkage maps were developed using SSR or restriction-site-associated DNA markers. There were three QTLs for “percentage of wilt leaves” in populations A, B and C, while populations A and B had three QTLs and population C had two QTLs for “days to wilt”. In population C, four QTLs were detected for Na^+^ concentration in the primary leaf. Among the F_2_ individuals in population C, 24% showed higher salt tolerance than both wild parents, suggesting that the salt tolerance of azuki beans can be further improved by combining the QTL alleles of the two wild relatives. The marker information would facilitate the transfer of salt tolerance alleles from “Tojinbaka” and “Ukushima” to azuki beans.

## 1. Introduction

The azuki bean (*Vigna angularis*) is one of the most important legumes in East Asia, particularly in Japan and China [1]. It is an important source of protein in this region and is cooked with rice in times of celebration. This bean has also been used as an ingredient for making traditional Japanese confectionary (called “Wagashi”) and is attracting attention to health-conscious consumers in the U.S.A. and Europe. To expand the production area of this crop, it is necessary to improve its ability to tolerate saline soil. Azuki bean is ranked in the most salt-susceptible group IV [2]. 

To improve the salt tolerance in azuki beans, identifying promising genes is necessary. In our study, we screened 69 azuki bean landraces together with cross-compatible wild relatives consisting of 150 accessions in 8 species [3]. The primary screening was performed in soil culture, followed by secondary screening in hydroponic culture, neither of which revealed a good source of salt tolerance from azuki bean landraces. In contrast, two accessions, *V. riukiuensis* “Tojinbaka” and *V. nakashimae* “Ukushima”, were identified as the most promising sources of salt tolerance, with each having different tolerance mechanisms. “Tojinbaka” accumulated Na^+^ throughout the whole plant body (Na^+^ includer), while “Ukushima” prevented Na^+^ accumulation in leaves by filtering Na^+^ in the roots and stems (Na^+^ excluder) [3,4]. In addition, “Ukushima” and “Tojinbaka” showed improved growth in a sea-water-damaged field from a tsunami in Fukushima, Japan, in which the local soybean cultivar, “Tachinagaha”, could not survive [3]. 

From DNA barcoding sequence analysis of rDNA-ITS and *atpB-rbcL* by Takahashi et al. [5], it was observed that *V. riukiuensis* and *V. nakashimae* are different species but are genetically closely related. *V. riukiuensis* is adapted to the coastal saline environment in the Ryukyu islands, Japan, whereas *V. nakashimae* is adapted to the northeastern region of East Asia and is mainly found in and around the Korean Peninsula. Among the *V. nakashimae* populations, some populations such as “Ukushima” adapted to the coastal saline environment and acquired salt tolerance [6]. *V. riukiuensis* and *V. nakashimae* are considered to have evolved from the common ancestral species *V. minima* [7]. *V. minima* originated in mainland Southeast Asia, and *V. riukiuensis* and *V. nakashimae* might have evolved from *V. minima* in the course of the expanding geographical distribution to East Asia. The time of divergence between *V. riukiuensis* and *V. nakashimae* was estimated to be only 60,000 years ago [2]. It is interesting that two genetically closely related species, “Tojinbaka” and “Ukushima”, have different salt tolerance mechanisms.

Genetic mapping is a powerful tool for detecting the loci involved in various biological processes [8]. The identification of genes that control morphological and physiological traits is crucial for understanding the mechanisms of adaptive evolution and for plant breeding. Simple sequence repeat (SSR) markers are highly reproducible, have codominant inheritance, are relatively abundant, show high levels of polymorphism and are easy to genotype, making them particularly useful for gene characterization and advantageous for mapping studies [9,10,11,12]. Additionally, a genome-wide microsatellite DNA marker database for three species within the *Vigna* species, including azduki bean, has been reported [13]. Recently, several SSR-based azuki bean linkage maps have been reported [14,15,16]. 

The advancement of next-generation sequencing (NGS) technologies has accelerated whole-genome sequencing and genotyping methods. Whole-genome sequences for 14 *Vigna* species have been constructed, including azuki bean (*V. angularis*) [17,18,19,20,21,22,23]. Genotyping using high-throughput NGS technologies reduces the complexity of genome analysis and allows for the development of a large number of SNP markers for any species of interest. Restriction-site-associated DNA sequencing (RAD-seq), including double-digested RAD-seq (ddRAD-seq), is one of the preferred NGS technologies for high-throughput genotyping. The RAD-seq method, first described by [24], has been used in many species without a reference genome, including the *Vigna* species *V. vexillata* (L.) A. Rich., *V. angularis* and *V. unguiculata* (L.) Walp [16,25,26,27]. While RAD-seq can be applied without a reference genome, a reference-guided RAD-seq approach can improve the accuracy of SNP detection. The high-quality *Vigna* genomes of 10 species including the azuki genome are available in the following database (VigGS: http://viggs.dna.affrc.go.jp/ (accessed on 1 March 2023)) [28]. Because the azuki genome is highly accurate, it can be used to generate draft genomes of closely related species or as a reference genome. The azuki genome can also be used for genetic analysis such as SNP marker detection in closely related wild species.

The objectives of this study were to (1) confirm the differences in salt tolerance mechanisms between “Tojinbaka” and “Ukushima”, (2) detect the genetic loci of salt tolerance in “Tojinbaka” and “Ukushima”, which could function on the azuki bean genetic background, and (3) investigate the possibility of achieving higher salt tolerance in azuki beans by pyramiding non-allelic QTLs from “Tojinbaka” and “Ukushima”.

## 2. Results 

### 2.1. Map Construction and Linkage Analysis of Population A and B Using SSR Markers

Among the 377 SSR markers screened in each population, 370 (99.2%) and 374 (98.1%) SSR markers amplified a fragment in populations A and B, respectively (Table 1). The percentage of SSR markers that amplified fragments ranged from 71.4% for cowpea SSR markers to 99.7% for azuki bean SSR markers. Among the 370 and 374 amplified SSR markers, 237 (62.9%) and 285 (75.6%) showed polymorphisms between parents in populations (A) and (B), respectively. The percentage of polymorphic SSR markers ranged from 37.5% for common bean SSR markers to 80.9% for azuki bean SSR markers (Table 1). 

In total, 193 (for population A) and 313 (for population B) clear polymorphic SSR products were mapped to linkage maps (Table 2, Appendix A). Markers could be assigned to 11 linkage groups (LGs), which coincide with the haploid chromosome number. The linkage maps spanned 560.2 cM for population A and 648.6 cM for population B in total length, with a mean distance between adjacent markers of 3.4 and 2.4 cM, respectively. In population A, the number of markers in the LGs ranged from 10 (LG7) to 41 (LG1) and the length of the LGs ranged from 31.8 cM (LG7) to 108.8 cM (LG1). In population B, the number of the markers ranged from 18 (LG9) to 59 (LG1) and the length of the LGs ranged from 35.5 cM (LG11) to 82.3 cM (LG1).

### 2.2. Map Construction and Linkage Analysis of Population C Using RAD Markers 

In total, 13,052,980 RAD-tag 85 bp reads of “Ukushima” were mapped onto the azuki genome to construct a custom pseudo-reference “Ukushima” genome (Appendix A). Further, 6,632,164 reads of “Tojinbaka” and 515,437,838 reads of F_2_ individuals were mapped onto the “Ukushima” genome. The average number of reads per F_2_ individual was 1,662,703. These mapped reads were used in SNP detection. In total, 74.1% (“Ukushima”), 62.0% (“Tojinbaka”) and 55.4% (F_2_) of the total number of reads were mapped onto the custom “Ukushima” genome. The mean coverage depths were 36.8 (“Ukushima”), 32.4 (“Tojinbaka”) and 17.4 (F_2_). The total number of detected SNPs was 25,350. Among them, 15,438 sites supported by Ukushima reads were reference-type alleles in Ukushima, and 5,469 of them were polymorphic in Tojinbaka. These sites were selected as bi-allelic RAD markers and were used in subsequent analysis. In F_2_, 2485 sites in Chr. 1 to 11 and 19 sites in scaffolds were available (Appendix A).

A total of 625 markers could be assigned to 11 linkage groups (LGs) (Table 2, Appendix A), which coincide with the haploid chromosome number. The LGs spanned 471.8 cM in total length, with a mean distance between adjacent markers of 0.9 cM. The number of markers in the LGs ranged from 55 (LG9) to 175 (LG1) and the length of the LGs ranged from 29.8 cM (LG3) to 67.8 cM (LG1) (Table 2). 

### 2.3. Measurement of Na^+^ Content in Leaves, Stems and Roots of Parental Materials

To confirm the differences in Na^+^ accumulation patterns among “Kyoto Dainagon”, “Tojinbaka” and “Ukushima”, the Na^+^ concentration in each of the following parts of the plants was measured: the 1st leaf to 5th (newest) leaf, the 1st to 3rd (newest) stem node and the roots. Yoshida et al. reported that Na^+^ concentration was measured, including the wilted leaves, and “Tojinbaka” was characterized as a “Na^+^ includer” and “Ukushima” as a “Na^+^ excluder” under 200 mM and 100 mM NaCl stress [3]. Noda et al. reported that Na^+^ allocation was measured in plants before wilting using radio-isotope ^22^Na^+^ with 100 mM NaCl treatment under artificial environmental conditions [4]. To confirm whether this phenomenon is reproduced in the greenhouse environment where the genetic analysis is performed, each parental plant was grown in a salt-treated environment and then divided into roots, stems and leaves to measure Na^+^ concentrations.

In general, the findings of Yoshida et al. [3] and Noda et el. [4] were confirmed. “Tojinbaka” was once again observed as a “Na^+^ includer”, while “Ukushima” showed a “Na^+^ excluder” type of Na^+^ accumulation pattern (Figure 1). In all of the plant parts, Na^+^ concentrations in “Tojinbaka” were higher than in “Ukushima”. However, it was revealed that “Tojinbaka” also had a Na^+^ filtering ability, since the Na^+^ concentration became lower in stems, particularly in the upper leaves, compared to the roots. On the contrary, “Kyoto Dainagon” did not show any Na^+^ filtering ability, even under mild salt stress, and accumulated more Na^+^ in the upper leaves. 

“Tojinbaka” continued to produce 4th and 5th leaves, even after a considerable amount of Na^+^ had accumulated in the leaves (Figure 1). On the contrary, “Ukushima” stopped growing despite lower Na^+^ concentrations in the leaves relative to “Tojinbaka”. 

### 2.4. QTL Analysis

#### 2.4.1. Positive and Negative Effects of “Tojinbaka” Alleles on Salt Tolerance in QTLs Detected from Population A (*V. angularis* “Kyoto Dainagon” × *V. riukiuensis* “Tojinbaka”)

In population A, 119 plants (62.6%) survived for one month under 100 mM NaCl conditions (Figure 2A), but only 24 F_2_ plants (12.6%) had no wilting leaves like “Tojinbaka” (Figure 2D). No F_2_ plants died earlier than “Kyoto Dainagon”. QTL analysis identified three QTLs for days to wilt (*qDtW1-1*, *qDtW2-1* and *qDtW8-1*) and three QTLs for percentage of wilt leaves (*qPWL1- 1*, *qPWL2-1* and *qPWL8-1*) in population A (Table 3). The QTLs for DtW and PWL on Chr 1 showed the highest LOD and PVE values. *qDtW1-1* could explain 15.0% of the variation in DtW, and *qPWL1-1* could explain 19.1% of the variation in PWL. QTLs on Chr 8 showed the second-highest LOD and PVE values. *qDtW8-1* could explain 12.0% of the variation and *qPWL8-1* could explain 15.9%. Each of the three QTLs for DtW and PWL detected at the adjacent region on the same chromosome (Chr) was considered to be the same allele because of the close mapped positions and the same direction of the allelic effect (Table 3, Figure 3). “Tojinbaka” alleles increased salt tolerance at QTLs detected on Chr 1 and Chr 2, while the “Tojinbaka” allele decreased salt tolerance at a QTL detected on Chr 8.

#### 2.4.2. “Ukushima” Allele Increases Salt Tolerance at QTLs Detected from Population B (*V. angularis* “Kyoto Dainagon” × *V. nakashimae* “Ukushima”)

In population B, 17 F_2_ plants (19.5%) could survive longer than “Ukushima” (Figure 2B), but only 6 F_2_ plants (6.9%) had no wilting leaves (Figure 2E). No F_2_ plants died earlier than “Kyoto Dainagon”. Three QTLs for days to wilt (*qDtW4-1*, *qDtW5-1* and *qDtW7-1*) and three QTLs for percentage of wilt leaves (*qPWL4-1*, *qPWL5-1* and *qPWL7-1*) were detected in population B (Table 3). The commonly obtained QTL on Chr 5 showed the highest LOD values. Based on the genome location and direction of the allele effect, *qDtW5-1* and *qPWL5-1* were considered to be the same “Ukushima” allele (Table 3, Figure 3). This QTL explained 15.8% of the variation in DtW and 17.5% of the variation in PWL. In all the QTLs detected in population B, “Ukushima” alleles increased salt tolerance, as expected, from its phenotype.

#### 2.4.3. “Tojinbaka” Allele Increases Salt Tolerance at QTLs Detected from Population C (*V. nakashimae* “Ukushima” × *V. riukiuensis* “Tojinbaka”)

To clarify the combined effects of salt tolerance QTLs from “Tojinbaka” and “Ukushima”, F_2_ population C was developed. As expected, most F_2_ plants from population C exhibited higher salt tolerance compared to plants from populations A and B (Figure 2). Therefore, the NaCl concentration was increased from 200 mM to 250 mM 25 days after salt treatment began. Under 250 mM salt stress conditions, the wild parent with the higher salt tolerance, “Tojinbaka”, could survive up to 55 days after salt treatment (Figure 2C). In addition, strong transgressive segregation was observed in population C. Among the 348 F_2_ segregating plants, 72 plants (24.0%) could survive longer than “Tojinbaka”. Strong transgressive segregation was also observed for PWL. One hundred F_2_ plants (32.8%) showed lower PWL compared to “Tojinbaka” (Figure 2F). Of these, 15 F_2_ plants (12.1%) showed no wilting leaves (PWL = 0).

Three QTLs for DtW (Chr 4, 5, 11) and two QTLs for PWL (Chr 4, 11) were detected in population C (Table 3). At all of the detected QTLs, “Tojinbaka” alleles increased salt tolerance. The QTLs detected on Chr 4 for DtW (*qDtW4-2*) and PWL (*qPWL4-2*) were positioned close to one another and were considered to be the same locus (Table 3, Figure 3). QTLs on Chr 4 showed the highest LOD and PVE values. This QTL explained 11.8% of the variation in DtW and 10.8% of the variation in PWL.

#### 2.4.4. Transgressive Segregation for Na^+^ Contents in Leaves in Population C (*V. nakashimae* “Ukushima” × *V. riukiuensis* “Tojinbaka”)

Since “Ukushima” is considered as a “Na^+^ excluder” and “Tojinbaka” is a “Na^+^ includer” (Figure 1), the Na^+^ concentration in the primary leaves of the F_2_ population was measured 17 days after the commencement of salt treatment for QTL analysis. The Na^+^ concentration in the primary leaf of “Ukushima” was still lower than 250 ppm at that time (Figure 4). However, “Ukushima” wilted completely (dying) three days after the Na^+^ concentration was measured (20 days after salt treatment). The Na^+^ concentration in the primary leaf of “Tojinbaka” was 2166 ppm, and this accession could survive up to 55 days under 250 mM saline conditions (Figure 2). In the F_2_ population, strong transgressive segregation was also observed in Na^+^ accumulation. There was a weak but significant correlation between Na^+^ concentration in primary leaves 17 days after salt treatment and days to wilt (*p* < 0.0001) in the F_2_ population (Appendix A). 

## 3. Discussion

In this research, three interspecific hybrids were developed, three linkage maps based on SSR or RAD markers were constructed using F_2:3_ or F_2_ derived from the combination of three parents, *Vigna angularis* “Kyoto Dainagon”, *V. riukiuensis* “Tojinbaka” and *V. nakashimae* “Ukushima”, and QTL analysis was conducted. All maps had 11 linkage groups (Table 2, Appendix A), which corresponded to the haploid chromosome number of these species [29]. Variant detection using RADseq allowed us to obtain many markers at once without prior screening, including SSR markers (Table 1). Although RAD may not be able to obtain enough polymorphic markers in very genetically close (e.g., between breeding cultivars) inbred populations within a species [30], we reconfirmed that it is a very powerful tool for large-scale genetic analysis in *Vigna* interspecific crosses. In contrast, small-scale marker analysis such as marker-assisted selection (MAS) requires markers that can be analyzed individually, such as SSR markers. Therefore, it is necessary to develop a small number of SNP markers that can be used in the future at a low cost for the MAS to introduce salt-tolerant QTLs in azuki beans.

The different Na^+^ allocation patterns in three accessions used in this study were previously reported by Yoshida et al. [3] and Noda et al. [4]. Yoshida et al. measured the Na^+^ concentrations in leaves, stems and roots under 200 mM salt concentrations for 16 days after salt treatment, in which “Kyoto Dainagon” completely died [3]. The authors found that “Ukushima” was the excluder type, with low Na^+^ concentrations from the roots to the leaves, while “Tojinbaka” was an includer with a high Na^+^ concentration from the root to the leaf. However, since 200 mM NaCl is fairly strong salt stress for azuki and related salt-tolerant species, Na^+^ distributions were observed in dying plants. Noda et al. investigated the radioisotope ^22^Na^+^ distribution under 100 mM salt stress conditions [4]. Three or six days after salt stress with radioactive ^22^Na^+^, they exposed whole plants to a storage phosphor screen for 24 h to visualize the ^22^Na^+^ distribution patterns in plants and then separated the root, stem and leaves and dried the samples to measure the ^22^Na^+^ radioactivity. At 3 days under 100 mM salt treatment, “Ukushima” and “Tojinbaka” mainly accumulated ^22^Na^+^ in the stem, while “Ukushima” maintained lower ^22^Na^+^ in the leaves compared with “Tojinbaka” [4]. Six days after 100 mM salt treatment, “Ukushima” showed ^22^Na^+^ filtering ability by the roots, resulting in the highest ^22^Na^+^ concentration in the roots and lowest ^22^Na^+^ concentration in the leaves. These results indicate that the Na^+^ allocation varies with salt concentration and salt stress period. In the present study, we cultivated these three accessions for 6 days under 100 mM salt treatment after 3 days under 50 mM salt treatment, dried them immediately and measured the Na^+^ concentration in each plant part based on the dry weight. Our results showed that the Na^+^ concentration was high and almost the same from the root to the first leaf in “Kyoto Dainagon”. This result indicates that “Kyoto Dainagon” does not have any Na^+^ filtering ability under 9 days of salt treatment (50 mM 3 days and 100 mM 6 days). “Ukushima” and “Tojinbaka” still retain the ability to filter Na^+^ from the stems and leaves upon 9 days of salt treatment, while the filtering ability was higher in “Ukushima”. This suggests that QTL analysis using populations A and B could detect salt-tolerant QTLs related to Na^+^ filtering ability from the stem and the leaf. Noda et al. [4] speculated that this mechanism of filtering and reducing the Na^+^ concentration in leaves might rely on Na^+^/K^+^ homeostasis in “Ukushima” and not in “Tojinbaka”. 

“Kyoto Dainagon” and “Ukushima” stopped growing after salt treatment started, whereas “Tojinbaka” could maintain growth even with Na^+^ accumulation in the leaves. The uppermost leaves of all accessions were the 3rd leaf at the beginning of salt treatment but only “Tojinbaka” could develop the 5th leaf when the Na^+^ concentration in each plant tissue was measured (9 days after salt treatment, Figure 1). This strongly suggests that “Tojinbaka” has a mechanism not only to filter Na^+^ allocation to the leaves, but also has a mechanism to detoxify Na^+^ in the leaves. In contrast, “Ukushima” may not have a mechanism to detoxify Na^+^ because it stopped growing, even though the Na^+^ concentration in leaves was lower compared to “Tojinbaka” (Figure 1). Thus, in population C, the detected QTLs might be related to filtering ability in reducing Na^+^ allocation to the leaves, as well as QTLs related to the Na^+^ detoxification mechanism. In addition, in population A, it is also possible to detect QTLs related to the detoxification mechanism. 

Generally, the detoxification mechanism is considered to be related to the vacuolar membrane transport systems, which lead to Na^+^ sequestration in vacuoles [31,32,33,34,35,36,37,38,39,40,41]. However, Noda et al. reported that there was no difference in the development and features of vacuoles between salt-sensitive *V. angularis* cv. “Erimoshouzu” and “Tojinbaka” after 3 days of 100 mM salt (NaCl) treatment with 5 kBq ^22^Na^+^ [42]. The only difference they could find between “Erimoshouzu” and “Tojinbaka” was that “Tojinbanka” contained well-developed starch granules in the chloroplast, whereas “Erimoshouzu” did not. In addition, ^22^Na^+^ was co-localized around starch granules developed in chloroplasts in the leaves of “Tojinbaka” and was not detected in other parts in the leaf cell. Therefore, they speculated that “Tojinbaka” has a Na^+^ trapping system using Na^+^-binding starch granules, which may be related to salt tolerance [42]. Common reed (*Phragmites australis*), a well-known salt-tolerant plant, also produced a large number of Na^+^-binding starch granules at the shoot base when salt stressed, and it was proposed as a novel salt tolerance mechanism not reported before [43]. The QTLs detected in population A might be related to genes involved in the formation of Na^+^-binding starch granules in the chloroplasts of the leaf cells, which could be part of the mechanism for salt tolerance in “Tojinbaka”. Interestingly, Noda et al. reported that “Ukushima” also contained starch granules in the chloroplasts [42]. However, they could not observe the co-localization of ^22^Na^+^ with starch granules in the chloroplasts of “Ukushima”. Therefore, they proposed the possibility that the starch granules in “Ukushima” may lack Na^+^ binding ability. Further studies are needed to confirm this hypothesis.

Four QTLs related to Na^+^ filtering were detected based on the Na+ concentration in the primary leaf in population C (Table 3). Among the four QTLs, the “Tojinbaka” alleles of QTLs (*qNa8-1* and *qNa11-1*) increased the Na^+^ concentration. In population B, no salt-tolerance-related QTL was detected in these two regions. In population A, salt-tolerance-related QTLs were detected near the *qNa8-1* region as *qDtW8-1* and *qPWL8-1*, and no polymorphic markers were present in the *qNa11-1* region (Figure 3). Thus, these two QTLs are likely due to the effect of the “Tojinbaka” allele on increasing Na^+^ concentration in the primary leaf. The other two QTLs of “Tojinbaka” alleles (*qNa2-1* and *qNa8-2*) decreased the Na^+^ concentrations in the primary leaf. This indicated that “Tojinbaka”, which was characterized as a “Na^+^ includer” in previous studies [3,4], had QTLs with opposite effects of suppressing Na^+^ accumulation in the leaves. Surprisingly, these two “Tojinbaka” alleles at these two QTLs with Na^+^ filtering effects (*qNa2-1* and *qNa8-2*) showed higher LOD (8.1 and 7.1) and PVE (11.1% and 9.7%) values (Table 3). In contrast, no QTLs were detected for the effect of the “Ukushima” allele to reduce the Na^+^ concentration in primary leaves. These results suggest that the effect of Na^+^ filtering genes in “Ukushima” might be small enough to be detected as QTLs or might be obscured by interactions with other QTLs. It is also possible that “Ukushima” and “Tojinbaka” have the same genes (QTLs) for Na^+^ filtering from leaves.

The “Tojinbaka” allele in the QTL region (*qNa8-1*) increased the Na^+^ concentration in population C and decreased the salt tolerance (*qDtW8-1* and *qPWL8-1*) in population A (Table 3, Figure 3). These three QTLs mapped around the 5 Mb region on Chr. 8, detected in both populations A and C, could be considered as the same locus and are thought to have decreased salt tolerance in population A due to increased Na^+^ in leaves by the “Tojinbaka” allele. In population A, a genetic interaction was observed between QTLs *qDtW8-1*/*qPWL8-1* and *qDtW1-1*/*qPWL1-1* (Appendix A). When the genotype of *qDtW8-1*/*qPWL8-1* has the “Tojinbaka” allele (BB), which is effective on increased Na^+^ accumulation in leaves, DtW dramatically decreased when the genotype of *qDtW1-1*/*qPWL1-1* has the “Kyoto Dainagon” allele (AA). However, even when the genotype of *qDtW8-1*/*qPWL8-1* has the “Tojinbaka” allele (BB), which is effective on increased Na^+^ accumulation in leaves, DtW does not decrease when the genotype of *qDtW1-1*/*qPWL1-1* carries the “Tojinbaka” allele (BB). This phenomenon suggests that the “Tojinbaka” allele of *qDtW1-1*/*qPWL1-1* might be involved in the detoxification of Na^+^ in the leaves. It is possible that these QTLs were involved in the Na^+^ trapping system using starch granules [42]. In population C, the “Tojinbaka” allele of *qNa8-1* had no significant effect on DtW and PWL but an increased Na^+^ concentration in leaves (Table 3, Figure 3). This suggests that other QTLs are more effective on Na^+^ detoxification than *qDtW1-1* in the genetic background of wild species with high Na^+^ filtering ability.

## 4. Materials and Methods

### 4.1. Parental Materials

Three accessions from three *Vigna* species were used in this study as parental materials: *V. angularis* (azuki bean cultivar “Kyoto Dainagon”, JP109685, salt-susceptible), *V. riukiuensis* (wild accession “Tojinbaka”, JP235833, salt-tolerant) and *V. nakashimae* (wild accession “Ukushima”, JP107879, salt-tolerant). “Tojinbaka” was collected from Ishigakijima island in Okinawa prefecture, Japan (https://www.gene.affrc.go.jp/databases-plant_search_detail_en.php?jp=235833 (accessed on 1 March 2023)). “Ukushima” was collected from Ukushima island in Nagasaki prefecture, Japan (https://www.gene.affrc.go.jp/databases-plant_search_detail_en.php?jp=107879 (accessed on 1 March 2023)). The habitats of these wild accessions are close to the seashore.

### 4.2. Three Interspecific Hybrid Populations

To develop genetic linkage maps and to conduct QTL analysis, three interspecific hybrid populations were developed: (A) *V. angularis* “Kyoto Dainagon” × *V. riukiuensis* “Tojinbaka”, (B) *V. angularis* “Kyoto Dainagon” × *V. nakashimae* “Ukushima” and (C) *V. nakashimae* “Ukushima” × *V. riukiuensis* “Tojinbaka”. In all cross combinations, an accession with higher salt tolerance was used as the male parent. For population A, the F_2_ and F_2:3_ population consisting of 190 individuals was used to develop a genetic linkage map and to phenotype salt tolerance. For population B, the F_2_ population consisting of 211 individuals was used to develop a genetic linkage map, and 87 F_2:3_ lines (with 1 to 10 plants per line) were used to phenotype salt tolerance. For population C, the F_2_ population consisting of 348 individuals was used to develop a genetic linkage map and to phenotype salt tolerance.

### 4.3. Growth Conditions and Salt Stress Treatments

The F_2_ plants and F_2:3_ lines of hybrid populations A and B were used for salt tolerance phenotyping, respectively. Seeds were sown in trays with Seramis clay granules (SERAMIS^®^) and were supported and kept in a growth chamber at 25 °C for 10 days. Each plant was transplanted to a hydroponic culture in a greenhouse of the Genetic Resources Center, Tsukuba, Japan (36°03′ N, 140°10′ E), under short-day conditions (10 h light/14 h dark). A piece of each plant’s leaves was sampled for DNA extraction. The hydroponic culture solution contained a diluted nutrient solution with a 1:1 ratio of Otsuka house No. 1 (1.5 g/L): Otsuka house No. 2 (1 g/L) (Otsuka Chemical Co., Osaka, Japan: N, P, K, Ca, and Mg = 18.6, 5.1, 8.6, 8.2 and 3.0 mEq/L, respectively). The final solution was adjusted to an EC of 100 mS/m with water. Three weeks after transplanting, NaCl treatment was initiated. NaCl concentration was 50 mM for the first four days and was increased by 50 mM at four-day intervals until a final concentration of 200 mM was reached. After that, the NaCl concentration was maintained at 200 mM for 16 days (until 28 days after commencement of salt treatment). Accessions that survived the salt treatment were grown under lower salt conditions after the experiment and were used for F_4_ seed production.

In the case of hybrid population C, the F_2_ seeds were sown and transplanted as described above. Three weeks after transplanting, NaCl treatment was initiated. NaCl concentration was 50 mM for the first three days and was increased by 50 mM at three-day intervals until a concentration of 200 mM was reached. After that, the NaCl concentration was maintained at 200 mM for 16 days (25 days after salt commencement of treatment). The primary leaves were sampled 17 days after salt treatment began, and Na^+^ concentration in the primary leaves was measured by the method later described in “*Na^+^ concentration measurement in parental lines*”. Potentially because both wild parents have high levels of salt tolerance, many F_2_ individuals were still healthy 25 days after salt treatment began. Therefore, NaCl concentration was further increased to 250 mM and was maintained at 250 mM until 76 days after commencement of salt treatment.

### 4.4. Measurement of Na^+^ Concentration in Parental Lines

To compare the pattern of Na***^+^*** accumulation in different plant parts, three seeds each from “Kyoto Dainagon”, “Tojinbaka” and Ukushima” were sown and transplanted to hydroponic solution, as described above. Three weeks after transplanting, NaCl treatment was initiated. NaCl concentration was 50 mM for the first four days and was increased to a final concentration of 100 mM. Six days after the NaCl concentration reached 100 mM (nine days after salt treatment began), the leaves, stems and roots were harvested separately. The leaves were further divided according to the stem nodes on which they grew, from the primary node up to the fifth node. The stems were also divided into three parts: (1) from the base of the stem to the node of the primary leaf, (2) from the first node to second node and (3) the remaining part of the stem above the second node. 

Measurement of Na^+^ concentration was conducted using the ion meter method [2], with slight modifications. The harvested plant samples with three replications were dried at 80 °C for 24 h and were ground into a powder. Roughly 20 mg of powder was collected into 2 cc tubes, and 700 µL of extraction solution (1 N ammonium acetate) was added and mixed at 90 °C for 1 h. After centrifugation at 7000 rpm (4400× *g*) for 5 min, 200 µL of supernatant was analyzed with a LAQUA twin Na^+^ meter (HORIBA Advanced Techno Co., Ltd. Kyoto, Japan). Before measurements, the LAQUA twin Na^+^ meter was calibrated using the corresponding calibration solution. All measurement procedures were conducted at room temperature (approx. 20 °C). 

### 4.5. Evaluation of Salt Tolerance

Two salt-tolerance-related traits were recorded for each plant or line in populations A and B, respectively. The first trait, called ‘Days to Wilt’ (*DtW*), recorded the number of days from salt treatment until the plant wilted. The second trait, known as ‘Percentage of Wilt Leaves’ (*PWL*), measured the proportion of leaves that had wilted, a symptom of the plant dying. *PWL* was visually scored using eight categories: 0% = normal healthy leaves, 15% = 1–15% of leaves wilted, 30% = 16–30% of leaves wilted, 45% = 31–45% of leaves wilted, 60% = 46–60% of leaves wilted, 75% = 61–75% of leaves wilted, 90% = 76–90% of leaves wilted and 100% ≥ 90% of leaves wilted or plant completely dead. *PWL* scores were recorded 19 days after salt treatment began. 

In addition to *DtW* and *PWL*, the Na^+^ concentration of primary leaves was measured in population C 17 days after NaCl treatment began (9 days after 200 mM). The method of Na^+^ measurement was the same as described above.

### 4.6. SSR Marker Analysis and Construction of Linkage Maps of Populations A and B

DNA was extracted from young leaf samples using the EZ1 DNA Tissue kit (QIAGEN, Valencia, CA, USA). The extracted DNA samples were used for PCR amplification using 330 azuki bean SSR primer sets [44], 40 common bean SSR primer sets [45,46] and 7 cowpea SSR primer sets [47]. 

PCR amplification was performed with a 5 μL reaction mixture containing 5 ng DNA template, 10x PCR buffer, 2 mM dNTP, 25 ng of each primer and 0.125 U Ex Taq polymerase enzyme. The thermal cycler protocol consisted of denaturation at 94 °C for 30 s, 40 cycles of 94 °C for 30 s, 55 °C for 60 s and 72 °C for 60 s, followed by 72 °C for 10 min cooling at 10 °C. PCR was performed in multiplex PCR reactions. Subsequently, PCR product size was analyzed on an ABI Prism 3100 genetic analyzer using GENEMAPPER ver. 3.0 (Applied Biosystems). The two linkage maps were constructed using AntMap [48].

### 4.7. Construction of V. nakashimae “Ukushima” Pseudo-Reference Genome for RADseq

To improve the mapping quality of reads from wild species to the azuki genome, a pseudo-custom *V. nakashimae* “Ukushima” genome was constructed. The raw RAD reads of “Ukushima” were first processed via the Trimmomatic-0.30 to remove the reads with sequencing adaptors and of low quality (phred quality < 5) [49]. Then, the processed reads were mapped onto the azuki genome (Vangularis_v1: http://viggs.dna.affrc.go.jp/ (accessed on 4 August 2015)) using Burrows–Wheeler Aligner (BWA) v.0.7.9a [50]. For local realignment and base quality score recalibration of the mapped reads, the RealignerTargetCreator and IndelRealigner tools from the GATK v3.0.0 software package [51] were applied. All variants were named using GATK with HaplotypeCaller. The genomic sequence of azuki was replaced with the variant of *V. nakashimae* using GATK with FastaAlternateReferenceMarker, and the result was used as a custom “Ukushima” pseudo-reference genome.

### 4.8. RADseq Analysis and Construction of Linkage Map for Population C

Genomic DNA was extracted from parents and 348 F_2_ plants from population C (*V. riukiuensis* × *V. nakashimae*) using the CTAB method [52]. Each genomic DNA sample (100–300 ng) from parents and F_2_ individuals was simultaneously digested with *BamHI-HF* and *NlaIII* and purified. Adaptor-1 and adaptor-2 were ligated to the digested DNA samples and purified. Adaptor-ligated biotinylated DNA samples were collected using streptavidin-coated magnetic beads (Dynabeads M270, Dynal). Adaptor-ligated DNA on the beads was amplified by PCR using Phusion High-Fidelity DNA polymerase (Thermo Fisher Scientific, Waltham, MA, USA) and the adapter primers. The size of the PCR-amplified fragments (smear between 200 and 1000 bp) was checked using electrophoresis in 1% agarose gel. The 96 purified PCR products were pooled and sequenced using the Illumina HiSeq2000 system. Sequence reads were mapped to custom pseudo-reference “Ukushima” genome, and local realignment and SNP detection were carried out as mentioned above. Error correction was performed based on the reference genome sequence. 

The genetic linkage group was constructed using R/qtl ver. 1.36.6 software [53]. For potential RAD markers, we selected bi-allelic markers based on parents” data, markers with genotyped plants more than 100 F_2_ individuals and dropped segregation distortion markers (chi-squared test, *p* < 10^−5^). Suitable values for recombination fractions and LOD scores were estimated using the “est.rf” command. After these steps, initial linkage group construction was carried out using the “Form Linkage Groups” command, with a maximum recombination fraction of 0.25 and a LOD threshold of 15. Robustness of linkage groups was checked using the “plot.rf” command. Marker order within linkage groups was estimated using a program called TMAP [54]. Some RAD markers mapped at loci less than 0.1 cM were not used for linkage map construction.

### 4.9. QTL Analysis

Single QTL analysis was run using the standard interval mapping method, as described in R/qtl [39]. The significance thresholds were calculated using 1000 permutation tests for each phenotype. Only those with an LOD score of three or more were adopted as QTLs in order to achieve a more stringent level. Interactions between QTLs and additional QTLs were examined using the “addint” and “addqtl” functions. The final multiple-QTL model fit was determined by drop-one-qtl analysis using the “fitqtl” function. 

To compare QTL positions across the three populations, the physical positions of SSR markers mapped on the linkage maps of the *V. angularis* × *V. riukiuensis* and *V. angularis* × *V. nakashimae* populations of the azuki reference genome sequence (Vangularis_v1: cultivar Shumari in VigGS: https://viggs.dna.affrc.go.jp/ (accessed on 18 June 2015) [29]) were determined by using BLASTN on the SSR primer sequences. The SSR, RAD markers and detected QTLs were distributed into 11 chromosomes according to their physical positions on the azuki reference genome.

## 5. Conclusions

This study demonstrated that the mechanisms underlying salt tolerance vary among azuki beans and their wild relatives. The salt-sensitive azuki bean *V. angularis* “Kyoto Dainagon” lacked Na^+^ filtering ability, while the salt-tolerant wild relatives *V. nakashimae* “Ukushima” and *V. riukiuensis* “Tojinbaka” were able to maintain lower Na^+^ concentrations in their leaves and had Na^+^ filtering ability. *V. nakashimae* had particularly strong Na^+^ filtering ability, but growth was stopped after salt treatment. In contrast, although *V. riukiuensis* accumulated Na^+^ in its leaves (about half to two-thirds concentration of azuki bean), growth was sustained. We developed three interspecific hybrid populations and identified 21 QTLs for salt tolerance. In population A (cross between azuki bean and *V. riukiuensis*), three QTLs were detected for DtW and PWL (*qDtW1-1*/*qPWL-1*, *qDtW2-1*/*qPWL2-1* and *qDtW8-1*/*qPWL8-1)*. In population B (cross between azuki bean and *V. nakashimae*), five QTLs were detected (*qDtW 4-1*, *qDtW5-1*/*qPWL5-1*, *qDtW 7-1*, *qPWL 4-1* and *qPWL7-1*). In population C (cross between *V. nakashimae* and *V. riukiuensis*), four QTLs were detected (*qDtW4-2*/*qPWL4-2*, *qDtW5-2*, *qDtW11-1* and *qPWL11-1*). Additionally, four QTLs (*qNa2-1*, *qNa8-1*, *qNa 8-2* and *qNa 11-1*) that were detected in the leaf Na^+^ concentration analysis performed only in population C did not match the location of QTLs for DtW and PWL in population C. However, *qDtW8-1*/*qPWL8-1* with the effect of the *V. riukiuensis* allele on reduced salt tolerance, which was detected in population A, was located on the same region as *qNa8-1*. This suggests that *qDtW8-1*/*qPWL8-1*/*qNa8-1* has an effect on reduced salt tolerance by promoting Na^+^ accumulation in the leaves. Additionally, a genetic interaction was observed between *qDtW8-1*/*qPWL8-1*/*qNa8-1* and *qDtW1-1*/*qPWL1-1*. This suggests that *qDtW8-1*/*qPWL8-1*/*qNa8-1* is associated with Na^+^ accumulation ability in the leaves in *V. riukiuensis*, and *qDtW1-1*/*qPWL1-1* is involved in the detoxification mechanism of Na^+^ that allows for growth under the presence of Na^+^ in the leaves. Furthermore, in population C, transgressive segregation was observed, indicating that the salt tolerance mechanisms in the two wild species might be different and that stronger salt tolerance could be acquired by combining genes conferring these mechanisms. Finally, this study not only provides useful marker information for subsequent fine mapping of candidate genes involved in salt tolerance and elucidating the genetic basis of Na^+^ filtering and detoxification mechanisms in the future but also contribute to developing super-salt-tolerant azuki beans by combining the salt-tolerant mechanisms of *V. riukiuensis* and *V. nakashimae*.

## Figures and Tables

**Figure 1 plants-12-01680-f001:**
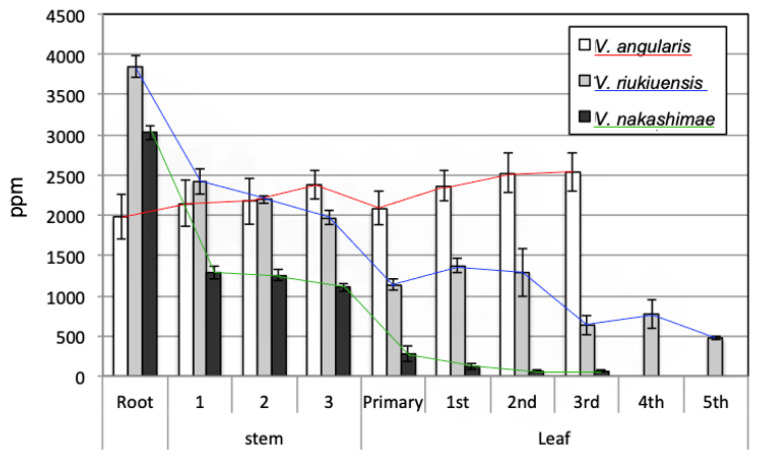
Na^+^ concentrations in different plant tissues of azuki bean “Kyoto Dainagon” and two salt-tolerant wild relatives, *V. riukiuensis* “Tojinbaka” and *V. nakashimae* “Ukushima”. Leaves were divided by stem node position. Stems were divided into three parts: 1 is base to primary node; “3” is top to two nodes from the end of stem; “2” is the middle part of stem other than 1 and 3. Samples were collected 6 days after 100 mM NaCl treatment (9 days after salt treatment). All leaves were still green at sampling. Error bars indicate standard deviations (*n* = 3). (ppm: parts per million).

**Figure 2 plants-12-01680-f002:**
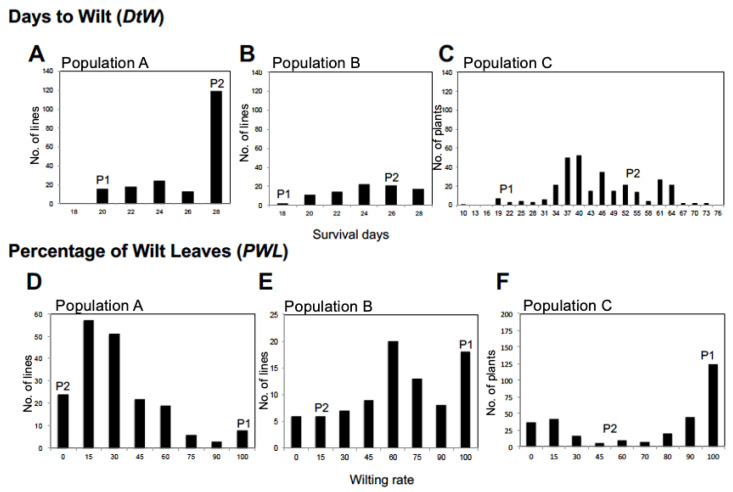
Frequency distribution of individuals for (upper: **A**–**C**) days to wilt (*DtW*) and (bottom: **D**–**F**) percentage of wilt leaves in the F_2:3_ segregating (**A**,**D**) population A; *V. angularis* (P1: “Kyoto Dainagon”) × *V. riukiuensis* (P2: “Tojinbaka”), (**B**,**E**) population B; *V. angularis* (P1: “Kyoto Dainagon”) × *V. nakashimae* (P2: “Ukushima”) and in the F_2:_ segregating (**C**,**F**) population C; *V. nakashimae* (P1: “Ukushima”) × *V. riukiuensis* (P2: “Tojinbaka”).

**Figure 3 plants-12-01680-f003:**
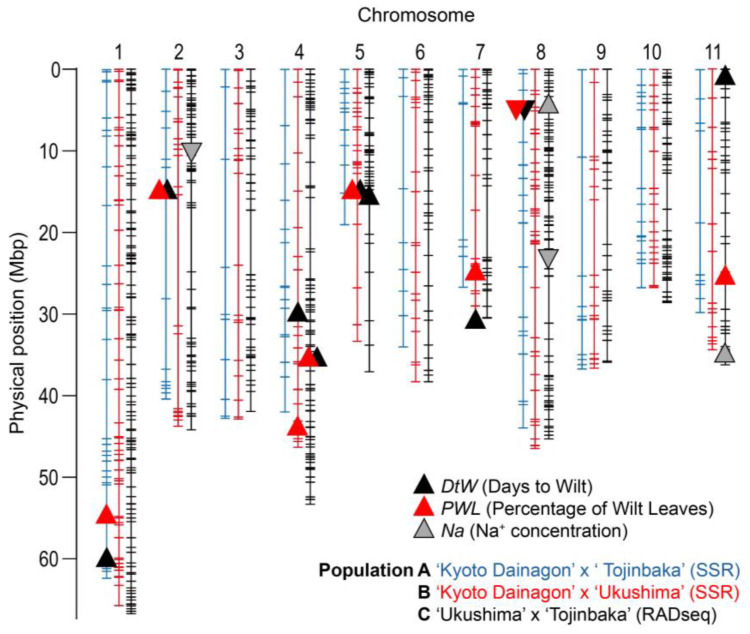
Comparative QTL locations detected from three populations, A: *V. angularis* “Kyoto Dainagon” × *V. riukiuensis* “Tojinbaka”), B: *V. angularis* “Kyoto Dainagon” × *V. nakashimae* “Ukushima” and C: *V. nakashimae* “Ukushima” × *V. riukiuensis* “Tojinbaka”. ▲ (upright triangles) indicate that the underlined strain showed increased salt tolerance or increased Na^+^ concentration. ▼ (inverted triangles) indicate the reverse effect direction.

**Figure 4 plants-12-01680-f004:**
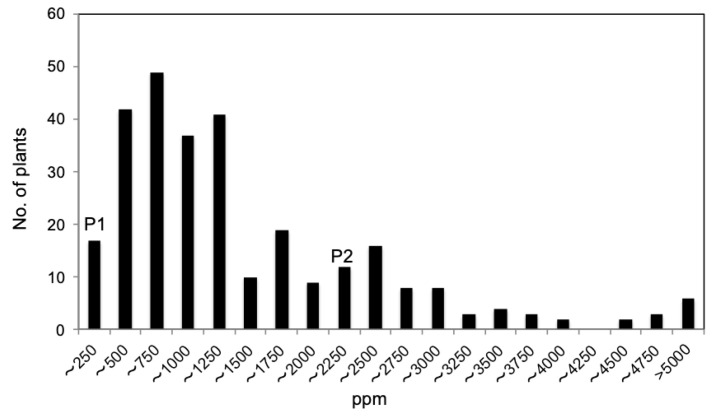
Frequency distribution of Na^+^ concentrations (ppm: perts per million) per dry weight of primary leaf in F_2_ population C and its parents. P1 and P2 represent the parent accession: *V. nakashimae* (P1: “Ukushima) and *V. riukiuensis* (P2: “Tojinbaka”).

**Table 1 plants-12-01680-t001:** Summary of SSR marker screening.

Population.	Marker Sources	Number of SSR Markers		
		Screened	Amplified (%)	Polymorphic (%)
**Population A** (*V. angularis* × *V. riukiuensis*)	Azuki bean	330	326 (98.8)	218 (66.1)
	Common bean	40	38 (95.0)	16 (40)
	Cowpea	7	6 (85.7)	3 (42.9)
	Total	377	370 (98.1)	237 (62.9)
**Population B** (*V. angularis* × *V. nakashimae*)	Azuki bean	330	329 (99.7)	267 (80.9)
	Common bean	40	40 (100)	15 (37.5)
	Cowpea	7	5 (71.4)	3 (42.9)
	Total	377	374 (99.2)	285 (75.6)

**Table 2 plants-12-01680-t002:** Characteristics of the linkage maps constructed for populations A, B and C.

Chr * ^1^	Population A			Population B			Population C		
	*V. angularis* × *V. riukiuensis* “Kyoto Dainagon” × “Tojinbaka”			*V. angularis* × *V. nakashimae* “Kyoto Dainagon” × “Ukushima”			*V. nakashimae* × *V. riukiuensis* “Ukushima” × “Tojinbaka”		
	SSR			SSR			RAD		
	Number of SSRs	Length (cM)	Average Marker Interval (cM)	Number of SSRs	Length (cM)	Average Marker Interval (cM)	Number of RAD-Tags	Length (cM)	Average Marker Interval (cM)
1	41	108.8	2.7	59	82.3	1.4	175	67.8	0.6
2	18	74.3	4.4	28	78.6	2.9	101	57.6	0.9
3	12	60.11	5.5	23	62.9	2.9	77	29.8	0.6
4	22	54.5	2.6	31	66.0	2.2	101	45.4	0.8
5	13	38.5	3.2	24	40.1	1.7	76	32.7	0.8
6	12	32.2	2.3	19	50.9	2.8	79	39.9	1.0
7	10	31.8	3.5	19	47.7	3.1	57	36.9	1.2
8	20	61.1	3.2	41	78.7	2.0	126	55.6	0.7
9	13	54.3	4.5	18	62.4	3.7	55	34.5	1.1
10	21	54.2	2.7	32	43.6	1.4	71	39.9	0.8
11	11	50.5	3.1	19	35.5	2.0	62	31.7	0.9
Total	193	560.2	3.4	313	648.6	2.4	625	471.8	0.9

* ^1^ Chromosomes corresponding to linkage groups.

**Table 3 plants-12-01680-t003:** Characteristics of salt-tolerance-related QTLs detected in populations A, B and C.

Trait	Pop * ^1^	Chr	QTL Name (Effect) * ^2^	Nearest Marker	cM * ^3^	Position * ^4^	Flanking Marker		Genetic Position (cM) * ^5^		Physical Position (bp) * ^6^		LOD	AE * ^7^	DE * ^8^	PVE * ^9^ (%)
							Start	End	Start	End	Start	End				
Days to Wilt (DtW)	A	1	*qDtW1-1*(*riu^+^*)	CEDG242	87.5	60,079,393	CEDG032	CEDG262	84.4	90.6	56,014,121	61,236,791	7.9	1.0	0.8	15.0
		2	*qDtW2-1*(*riu^+^*)	CEDG261	40.4	15,586,067	CEDG100a	CEDG237	37.3	45.4	14,608,533	31,635,569	3.9	0.8	−0.4	7.2
		8	*qDtW8-1*(*riu^−^*)	PV-atcc001	5.0	5,178,308	CEDG082	CEDG230	2.1	22.3	3,163,061	11,545,276	6.4	−1.0	1.2	12.0
	B	4	*qDtW4-1*(*nak^+^*)	CEDC0131	33.3	30,554,067	CEDG084	BM146b	29.3	39.0	24,427,507	36,066,076	3.0	0.8	2.0	9.6
		5	*qDtW5-1*(*nak^+^*)	CEDG159b	37.0	14,493,626	CEDG008	BMd-12	31.1	38.3	11,712,796	16,853,225	4.7	1.4	1.3	15.8
		7	*qDtW7-1*(*nak^+^*)	CEDG279	35.5	35,335,774	CEDG273	CEDG279	30.6	35.5	34,238,882	35,335,774	3.0	1.2	−0.3	9.8
	C	4	*qDtW4-2* (*riu^+^*)	4_36004406	34.2	36,004,406	4_33167677	4_39320776	29.3	39.6	33,167,677	39,320,776	8.7	3.6	5.2	11.8
		5	*qDtW5-2*(*riu^+^*)	5_15717934	30.8	15,717,934	5_12972702	5_21478497	25.7	32.1	12,972,702	21,478,497	4.0	2.5	2.7	5.2
		11	*qDtW11-1*(*riu^+^*)	11_1337709	1.3	1,337,709	11_483319	11_6994697	0.0	6.7	483,319	6,994,697	3.0	2.4	3.6	3.8
Percentage of Wilt Leaves (PWL)	A	1	*qPWL1-1*(*riu^+^*)	BM181	85.0	55,004,523	CEDG057	CEDG229	76.0	86.3	50,637,355	60,742,976	9.5	−0.7	−0.8	19.1
		2	*qPWL2-1*(*riu^+^*)	CEDG261	40.4	15,586,067	CEDG100a	CEDG237	37.3	45.4	10,787,607	31,635,668	4.4	−1.1	−0.2	8.2
		8	*qPWL8-1*(*riu^−^*)	PV-atcc001	5.0	5,178,308	CEDG082	CEDG230	2.1	22.3	3,163,061	11,545,276	8.1	0.8	−0.6	15.9
	B	4	*qPWL4-1*(*nak^+^*)	CEDG292	63.9	43,496,022	CEDG062	CEDG036	58.4	66.1	41,709,579	48,786,346	3.5	−1.4	−1.1	11.1
		5	*qPWL5-1*(*nak^+^*)	CEDG159b	37.0	14,493,626	CEDG008	BMd-12	31.1	38.3	11,712,796	16,853,062	6.9	−1.4	−1.1	17.5
		7	*qPWL7-1*(*nak^+^*)	CEDG203	42.0	25,558,505	CEDG258	CEDG085	29.6	47.7	8,383,454	30,376,496	5.9	−1.2	−0.7	20.1
	C	4	*qPWL4-2*(*riu^+^*)	4_36004406	34.2	36,004,406	4_33167677	4_39320776	29.3	39.6	33,167,677	39,320,776	7.5	−1.0	−1.3	10.8
		11	*qPWL11-1*(*riu^+^*)	11_35006063	24.7	29,906,300	11_32017713	11_36707994	19.8	31.7	32,017,713	36,707,994	3.0	−0.6	0.8	4.2
Na^+^ concentration (Na) *^10^	C	2	*qNa2-1*(*riu^−^*)	2_9439413	25.0	9,439,413	2_8179808	2_12077785	20.9	30.7	8,179,808	12,077,785	8.1	−94.4	−9.3	11.1
		8	*qNa8-1*(*riu^+^*)	8_4742809	7.5	4,742,809	8_4300670	8_7708714	6.2	11.1	4,300,670	7,708,714	4.6	74.0	−74.9	6.1
		8	*qNa8-2*(*riu^−^*)	8_23203914	37.2	23,203,914	8_22256774	8_35392038	36.8	39.4	22,256,774	35,392,038	7.1	−53.5	−47.6	9.7
		11	*qNa11-1*(*riu^+^*)	11_35676919	27.1	35,676,919	11_35339431	11_36707994	25.7	31.7	35,339,431	36,707,994	5.2	81.1	43.4	7.0

* ^1^ Population (A) *V. angularis* “Kyoto Dainagon” × *V. riukiuensis* “Tojinbaka”, (B) *V. angularis* “Kyoto Dainagon” × *V. nakashimae* “Ukushima” and (C) *V. nakashimae* “Ukushima” × *V. riukiuensis* “Tojinbaka”. * ^2^ QTL names (effect) consists of “trait” + “chromosome” + “QTL number in the chromosome” (“pollen parents name”, with “+” indicating that the allele of underlined strain increase salt tolerance [or Na^+^ concentration in primery leaf] and “-“indicating that it decreases it). * ^3^ The peak position of LOD score. * ^4^ The physical position of nearest marker on Azuki genome (Vangularis_v1.a1). * ^5^ The genetic position of flanking markers. * ^6^ The physical position of flanking markers on Azuki genome. * ^7^ AE: additive effect; positive values indicate that positive allele is derived from pollen parents (underlined above), while negative value indicates that positive allele is derived from maternal parents. A positive value of AE in DtW indicates an increase in salt tolerance, while a negative value of PWD improves salt tolerance. * ^8^ DE: dominance effect. * ^9^ PVE: percentage of the total phenotypic variation explained by each QTL. * ^10^ Na^+^ concentration in primary leaf.

## Data Availability

Data supporting the reported results are available on request from the corresponding author.

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
