# Peer review of "Unique Salt-Tolerance-Related QTLs, Evolved in Vigna riukiuensis (Na+ Includer) and V. nakashimae (Na+ Excluder), Shed Light on the Development of Super-Salt-Tolerant Azuki Bean (V. angularis) Cultivars"

_plants, 2023, doi:10.3390/plants12081680_

Round 1
Reviewer 1 Report
The manuscript titled "QTLs for salt tolerance in Vigna riukiuensis and Vigna nakashimae, close wild relatives of adzuki bean (Vigna angularis)" describes a study in which the authors used three mapping populations to identify QTLs for salt tolerance. Several alleles enhancing salt tolerance were identified in wild relatives of adzuki bean. The most interesting finding is that strong transgressive segregation was observed in population C (V. nakashimae “Ukushima” × V. riukiuensis “Tojinbaka”), suggesting the feasibility of improving the salt tolerance of adzuki bean by combining the salt-tolerant alleles from “Tojinbaka” and “Ukushima”. This manuscript is well-organized, written in a forward and to-the-point manner, and is worth considering for publication after addressing the comments below appropriately.
1. Figure 1: Please provide the unit of the vertical axis (Na+ concentration).
2. Line 209 (Figure 2): Please clarify the generation of the populations. According to the description in the Materials and Methods section (line 400), population A should be an F2 population. If an F2:3 segregating population is used, the vertical axis should be "No. of lines" instead of "No. of individuals".
3. Line 268: The meaning of "salt tolerance" is unclear. Does it mean "increased DtW value" and "decreased PWL value"? I am confused with the triangle directions shown in Figure 3 and the additive effect directions that were given in Table 3.
4. Line 282: There is no description of the QTL results for the Na+ concentration in the Results section.
5. Line 284: The legend of Figure 4 should be "Frequency distribution of Na+ concentration..."
6. Line 403: If F2:3 lines were used for salt tolerance phenotyping, more details about the phenotyping method should be provided, such as the number of plants per F2:3 line, etc.
7. There are numerous editorial errors throughout the manuscript, such as in line 177, where "n=3" should be replaced with "n = 3" and "n" should be italicized. In line 281, where "p=0.0001" should be replaced with "p = 0.0001" and "p" should be italicized. In line 365, where "5Mb" should be replaced with "5 Mb". In line 521, where "1000" should be replaced with "1,000".
Author Response
Thank you for your valuable comments and suggestions on our manuscript. We appreciate the time and effort you have put into reviewing our work and providing us with constructive feedback.
Comments and Responses:
- Figure 1: Please provide the unit of the vertical axis (Na+ concentration).
We have added the unit (ppm) of the vertical axis in Figure 1.
- Line 209 (Figure 2): Please clarify the generation of the populations. According to the description in the Materials and Methods section (line 400), population A should be an F2 population. If an F2:3 segregating population is used, the vertical axis should be "No. of lines" instead of "No. of individuals".
- Line 268: The meaning of "salt tolerance" is unclear. Does it mean "increased DtW value" and "decreased PWL value"? I am confused with the triangle directions shown in Figure 3 and the additive effect directions that were given in Table 3.
We have added an explanation in Table 3 for clarity (Line 264).
- Line 282: There is no description of the QTL results for the Na+ concentration in the Results section.
We have added the subsection title “2.4.4 Transgressive segregation for Na+ contents in leaves in population C (V. nakashimae “Ukushima” × V. riukiuensis “Tojinbaka”)” in Line 278.
- Line 284: The legend of Figure 4 should be "Frequency distribution of Na+ concentration..."
We have corrected the legend of Figure4. (Line 295)
- Line 403: If F2:3 lines were used for salt tolerance phenotyping, more details about the phenotyping method should be provided, such as the number of plants per F2:3 line, etc.
We have added number of plants per lines in Line 442.
- There are numerous editorial errors throughout the manuscript, such as in line 177, where "n=3" should be replaced with "n = 3" and "n" should be italicized. In line 281, where "p=0.0001" should be replaced with "p = 0.0001" and "p" should be italicized. In line 365, where "5Mb" should be replaced with "5 Mb". In line 521, where "1000" should be replaced with "1,000".
Thank you for pointing that out. All have been corrected. (Line 193, 290, 549)
Reviewer 2 Report
Genetic biodiversity of crops and it's wild relatives is important for to enhance food crops performance against abiotic stress tolerance. In present study authors have studied, two closely related wild species of the traditional East Asian legume crop, Adzuki bean (Vigna angularis), V. riukiuensis “Tojinbaka,” and V. nakashimae “Ukushima” and found to have much higher levels of salt tolerance than adzuki bean. Using SSR or restriction-site associated DNA markers, the authors developed linkage maps. The study showed large number of F2 individuals in population with higher salt tolerance when compared to both wild parents. It indicated that salt tolerance of adzuki bean was improved by combining the QTL alleles of the two wild relatives.
The study provide significant findings and useful for agronomic crop studies.
Minor comments:
1. Headlines of each result’s sub-sections need to be rewritten. Please write key significant finding as a heading for these sections.
2. The introduction needs to be expanded. Authors may consider following studies for introduction on molecular markers.
3. Multiplex molecular marker-assisted analysis of significant pathogens of cotton (Gossypium sp.), 2022; Biocatalysis and Agricultural Biotechnology https://doi.org/10.1016/j.bcab.2022.102557; Assessment of genetic diversity and volatile content of commercially grown banana (Musa spp.) cultivars, Hinge et al., Scientific Reports, 2022; https://doi.org/10.1038/s41598-022-11992-1 (Banana); Microsatellite analysis to differentiate clones of Thompson seedless grapevine, Upadhyay et al., 2010, Ind Journal of Horticulture, Volume 67 Issue 2 Pages 260-263.
4. Discussion is too broad need to be focused and concise.
5. In conclusions provide significant results and future implications of this study.
6. Figure 4. please elaborate the caption and explain the results in captions for easy understanding.
Author Response
Thank you for your valuable comments and suggestions on our manuscript. We appreciate the time and effort you have put into reviewing our work and providing us with constructive feedback.
Comments and responses
- Headlines of each result’s sub-sections need to be rewritten. Please write key significant finding as a heading for these sections.
As you pointed out, there was insufficient information in the sub-sections, so we have added key significant finding for headlines of sub-sections.
- The introduction needs to be expanded. Authors may consider following studies for introduction on molecular markers.
(1) Multiplex molecular marker-assisted analysis of significant pathogens of cotton (Gossypium sp.), 2022; Biocatalysis and Agricultural Biotechnology https://doi.org/10.1016/j.bcab.2022.102557; (2) Assessment of genetic diversity and volatile content of commercially grown banana (Musa spp.) cultivars, Hinge et al., Scientific Reports, 2022; https://doi.org/10.1038/s41598-022-11992-1 (Banana);
(3) Microsatellite analysis to differentiate clones of Thompson seedless grapevine, Upadhyay et al., 2010, Ind Journal of Horticulture, Volume 67 Issue 2 Pages 260-263.
We appreciate your suggestion to include references to three papers in our introduction. However, we would like to clarify that these papers are not directly related to the topic of our study, which is focused on marker analysis.
While we acknowledge that the papers you referenced may contain relevant information, their content is not in alignment with the scope of our research. Therefore, we did not include them in the introduction.
As you noted, we did not include an introduction on marker analysis. Therefore, we have added an introduction on marker analysis in azuki bean and its wild relatives.
- Discussion is too broad need to be focused and concise.
As you pointed out, discussion is too broad, so we have removed one paragraph.
Last month, Noda et al reported that the possibility of Na-trapping system by starch granules for salt tolerance in V. riukiuensis. This paper provides new insights as a candidate QTL for the detoxification shown in this paper. Therefore, it has been added to the discussion.
Reference: Noda et al., 2023
chrome-extension://efaidnbmnnnibpcajpcglclefindmkaj/https://assets.researchsquare.com/files/rs-2572700/v1/66ee9d61-e513-484b-ab04-e8f0ecd3c0d7.pdf?c=1677278406
- In conclusions provide significant results and future implications of this study.
We rewrite conclusion’s part which provide significant results and future implications of this study.
- Figure 4. please elaborate the caption and explain the results in captions for easy understanding.
The caption was difficult to understand, so we have corrected it for clarity.
Reviewer 3 Report
The research article aims to provide valuable information about QTLs for salt tolerance in close relatives of vigna angularis. However, there are few major concerns in the article to be addressed to improve it.
Comment: It is suggested to make the title catchy and appealing as the existing title merely looks methodological.
Comment: In line 36, the sentence “Linkage maps were drawn using SSR or…” can be changed to “linkage maps were developed using…”
Comment: In line 37, rewrite the sentence “Two or three QTLs for “percentage of wilt leaves” while mentioning the exact number of QTLs
Comment: Line 32, Rewrite the sentence “significant number of F2 individuals in population” with proper choice of words
Comment: Line 52-53, Rewrite the sentence “and is attracting attention of western countries”
Comment: Line 56, the introduction should be written more as an extract of the studies to support the background of current research instead of mentioning the methodology and results from the references.
Comment: Line 68, Rewrite the sentence “According to the molecular phylogenetic analysis by..”
Comment: Add reference for the line 73.
Comment: Line103, Subtitle can be changed to “Linkage analysis and Map construction”
Comment: Line 148, the results in the section titled "Confirmation of Na+ accumulation pattern of parental materials" should be rewritten as it is unclear and overly confusing.
Comment: Line 178, the writing appears to be confusing as the numerical terms are repeatedly used. Rewrite with proper choice of words.
Comment: Line 271, mention the Na ion accumulation results of population C under specific subtitle as they start abruptly after the QTL analysis parts.
Comment: More references should be included in the discussion of the results because only two or three are cited frequently throughout the paper.
Comment: Line 387, the salt tolerance trait of Kyoto Dainagon can be mentioned along with cultivar information in bracket.
Comment: Line 390, if the mentioned cultivars have been collected from the wild it is suggested to confirm them through molecular tools like DNA barcoding using MATK or RbCL genes.
Comment Line 399: Pollen parent can be replaced using male parent/donor.
Comment 400: While all the F2 individuals from the population A and C were used for the phenotyping, why only 87 F2:3 from the population B were used?
Comments: Line 407, repeated lines appear in section "Growth Conditions and Salt Stress Treatments," which requires revision.
Comment: Line 408, Above it is mentioned that F2 lines of population A and B were used for phenotyping.
Comment: Line 433, “Na+ concentration measurement in parental lines” can be changed to “Measurement of Na+ concentration in the parental lines”.
Comment: Line 438-443, the description of the sample collection method is quite ambiguous; revise the section to make it clearer.
Comment: Line 455-456, rewrite the sentence “These were: (1) number of days from salt treatment to wilt (DtW), and (2) percentage of wilt leaves (PWL). Wilt is a symptom of the plant dying require”.
Author Response
Thank you for your valuable comments and suggestions on our manuscript. We appreciate the time and effort you have put into reviewing our work and providing us with constructive feedback.
Comments and Responses
Comment: It is suggested to make the title catchy and appealing as the existing title merely looks methodological.
As you pointed out, the title was methodological, so we changed the title to “QTLs in wild species (Vigna riukiuensis and V. nakashimae) improves salt tolerant of cultivated azuki bean (V. angularis)”
Comment: In line 36, the sentence “Linkage maps were drawn using SSR or…” can be changed to “linkage maps were developed using…”
Thank you for pointing this out. We have corrected the sentence you pointed out.
Comment: In line 37, rewrite the sentence “Two or three QTLs for “percentage of wilt leaves” while mentioning the exact number of QTLs
We rewrited the sentence and adding the exact number of QTLs.
Comment: Line 32, Rewrite the sentence “significant number of F2 individuals in population” with proper choice of words
L39: We replaced “significant number” in this sentence with the exact percentage (24%) of F2 individuals. 
Comment: Line 52-53, Rewrite the sentence “and is attracting attention of western countries”
We replaced the sentence from “and is attracting attention of western countries” to “and is attraction attention in the U.S. and Europe health-conscious consumers”.
(Reference: https://sp.m.jiji.com/english/show/23946)
Comment: Line 56, the introduction should be written more as an extract of the studies to support the background of current research instead of mentioning the methodology and results from the references.
This study based on our previous work. We have modified it by making the subject matter ours.
Comment: Line 68, Rewrite the sentence “According to the molecular phylogenetic analysis by..”
We replaced the sentence from “According to the molecular phylogenetic analysis by..” to “DNA barcoding sequence analysis of rDNA-ITS and atpB-rbcL by..”.
Comment: Add reference for the line 73.
We added reference [6] for the line 76 in this version.
Comment: Line103, Subtitle can be changed to “Linkage analysis and Map construction” .
We changed the subtitle to “Linkage analysis and Map construction of population A and B using SSR marker” in Line 115. Accordingly, we changed the subtitle to “Linkage analysis and Map construction of population C and using RAD marker” in Line 138-9.
Comment: Line 148, the results in the section titled "Confirmation of Na+ accumulation pattern of parental materials" should be rewritten as it is unclear and overly confusing. .
We replaced the section title from “Confirmation of Na+ accumulation pattern of parental materials” to “Measurement of Na+ content in leaves, stems and roots of parental materials”
.
Comment: Line 178, the writing appears to be confusing as the numerical terms are repeatedly used. Rewrite with proper choice of words. .
We have removed some of the text and modified it to make it less confusing.
Comment: Line 271, mention the Na ion accumulation results of population C under specific subtitle as they start abruptly after the QTL analysis parts. .
We added new subtitle (2.4.4) after QTL analysis parts in L278-9.
Comment: More references should be included in the discussion of the results because only two or three are cited frequently throughout the paper. .
We added some references and discussion.
Comment: Line 387, the salt tolerance trait of Kyoto Dainagon can be mentioned along with cultivar information in bracket. .
We added “salt-susceptible” and “salt-tolerant” with cultivar information in bracket.
Comment: Line 390, if the mentioned cultivars have been collected from the wild it is suggested to confirm them through molecular tools like DNA barcoding using MATK or RbCL genes. .
“Kyoto Dainagon” was brought to Genebank from adzuki bead breeding site. And ”Ukushima” and “Tojinbaks” is not cultivar, but is wild species. It is not cultivated at all. Since the phenotypes are distinctly different, there is no need for confirmation by DNA barcoding.
Comment Line 399: Pollen parent can be replaced using male parent/donor. .
We replated from “pollen parent” to “make parent”.
Comment 400: While all the F2 individuals from the population A and C were used for the phenotyping, why only 87 F2:3 from the population B were used? .
In population C, the parents, V. nakashimae and V. riukiuensis are genetically very closely related, sterility rarely occurred, and a large number of seeds could be obtained. On the other hand, the population A and B, V. angularis and wild relatives (V. ankashimae and V. riukiuesis), were genetically more distant, resulting in frequent sterility and low seed production despite being able to mate. In particular, in population B, the F2 generation exhibited strong sterility, with only 83 F3 seed from F2 211 plants.
Comments: Line 407, repeated lines appear in section "Growth Conditions and Salt Stress Treatments," which requires revision. .
This is not a repetition, but rather a separate list of growing conditions for population [A,B] and [C]. It has been revised to eliminate the line break (line 455) and to make populations [A,B] and [C] their own paragraphs in Line 460.
Comment: Line 408, Above it is mentioned that F2 lines of popelulation A and B were used for phenotyping.
We didn’t performed phenotyping in F2 generation due to low seed production in population A and B. We used F2 generation only for propagation of F3 generation.
Comment: Line 433, “Na+ concentration measurement in parental lines” can be changed to “Measurement of Na+ concentration in the parental lines”.
We changed it to “Measurement of Na+ concentration in the parental lines” in Line 472.
Comment: Line 438-443, the description of the sample collection method is quite ambiguous; revise the section to make it clearer.
We have rewritten the sentence for clarity in line 451 and 460.
Comment: Line 455-456, rewrite the sentence “These were: (1) number of days from salt treatment to wilt (DtW), and (2) percentage of wilt leaves (PWL). Wilt is a symptom of the plant dying require”.
We have rewritten the sentence for clarity in line 481-5.
Round 2
Reviewer 3 Report
The revised version of the manuscript appears to have adequately addressed all of the concerns that I previously raised. Based on this, I am pleased to recommend acceptance of the manuscript.